# Special Anaesthetic Considerations for Brain Tumour Surgery in Children

**DOI:** 10.3390/children9101539

**Published:** 2022-10-09

**Authors:** Sandra Jeker, Maria Julia Beck, Thomas O. Erb

**Affiliations:** 1Department of Pediatric Anesthesia, University Children’s Hospital Basel (UKBB), 4056 Basel, Switzerland; 2Department of Pediatric Anesthesia, University of Basel, 4001 Basel, Switzerland

**Keywords:** paediatric anaesthesia, brain tumour, neurosurgery, cerebrovascular pediatric physiology, special considerations, pain management

## Abstract

Brain tumours are among the most common neoplasm in children. Therefore, paediatric anaesthesiologists face the challenge of neurosurgical interventions in all age groups. To minimize perioperative mortality and morbidity, a comprehensive understanding of age-dependent differences in anatomy and cerebrovascular physiology is a mandatory prerequisite. Advances in subspeciality training in paediatric neurosurgery and paediatric anaesthesia may improve clinical outcomes and advance communication between the teams.

## 1. Introduction

Paediatric anaesthesiologists face the unique challenge of treating children along the entire developmental spectrum from birth to adolescence. In order to satisfy the overall needs of this diverse population, the physiologic, anatomic, and psychologic differences of all age groups must be known and need to be addressed. As neurosurgical techniques evolve and interventions advance to ever more sophisticated methods, it is mandatory for paediatric anaesthesiologists and neurosurgeons alike to know the age-dependent differences of their patients, with the aim of reducing perioperative morbidity and mortality [1,2].

Additionally, the paediatric anaesthesiologist does not only communicate with the paediatric patient in their stage of cognitive development and anxiety, but also with the legal guardian with her or his own fears and concerns. In children, a condition in need of neurosurgical treatment challenges legal guardians with a high level of stress and uncertainty. Considerate communication skills and a coherent flow of information are crucial to empower the legal guardians to take on a supportive role in the child’s peri- and postoperative wellbeing.

Brain tumours are the most common solid neoplasms in children and are second only to leukaemia in all paediatric malignancies [3]. Histology and location, as well as physiology, are significantly different from those in adult patients. Paediatric brain tumour surgery can be high risk for various reasons, including the existence of large tumours, complex midline tumours, and posterior fossa tumours.

Anaesthesia for paediatric craniotomies can neither be separated from the general principles of paediatric anaesthesia nor from the fundamental principles of neuro-anaesthesia. The major challenges of this surgery are haemorrhagic risks, air embolism, hemodynamic instability, the management of cerebral blood flow, and intracranial hypertension, with the overall goal of preserving neurological function, hemodynamic stability, oxygen carrying capacity, and perfusion of all vital organs.

The goal of this review is to describe age-dependent differences in anatomy and cerebrovascular physiology in children undergoing craniotomy, with the aim to guide and optimize anaesthesia management in order to ensure maximal patient safety throughout the perioperative period [4].

## 2. Neurological Physiology

Precise data about neurophysiological values in the different paediatric age groups is limited. Normal ranges and treatment goals are frequently extrapolated from adult or animal data and very often based on expert consensus. Age-dependent differences influence the anaesthetic approach to the paediatric patient undergoing craniotomies.

The intracranial space contains three compartments: brain tissue (80–85%), cerebrospinal fluid (10%), and blood (5–10%). Cerebrospinal fluid (CSF) is steadily produced by the choroid plexus, circulates through the subarachnoid space and ventricles and is resorbed by the arachnoid granulations. Blood is mainly pooled in the venous sinuses and pial veins. Of note, the arterial system accounts for only around 25% of cerebral blood volume. Since the skull is a rigid structure, the total volume of the three compartments is fixed in children and exerts a certain intracranial pressure (ICP).

An increase in volume (e.g. tumour, hydrocephalus, abscess, bleeding) in either one of the intracranial compartments will result in an increased ICP and trigger compensatory mechanisms to reduce the volume of the remaining compartments. The extent, speed and duration of the physiological compensatory mechanisms is limited. Before cranial suture fusion, gradual decompression can additionally occur from an increase in skull size and can therefore mask classic symptoms of elevated ICP. Once intracranial compensation is exhausted (Figure 1), additional increases in intracranial volume result in a dramatic rise in ICP limiting cerebral blood flow (CBF) and calling for immediate intervention ([5], Figure 1).

The cerebral blood flow (CBF) and metabolism varies with age. Global CBF is lower in term and premature infants and gradually increases in infants and children aged from 6 months to 3 years compared to adults (see Table 1 below, [6]). Additionally, the pattern of regional blood flow changes with age, with a markedly higher rCBF in the grey matter and a lack of frontal predominance when compared to adults [7].

Cerebral metabolic rate for oxygen (CMRO_2_) is one of the determinants of CBF and is closely linked through all age groups, as supply and demand are closely linked. In neonates the CMRO_2_ is reduced with a relative tolerance to hypoxia, then doubling during early childhood and finally reaching somewhat lower adult levels. Furthermore, brain metabolism may directly regulate blood flow locally through chemical factors such as oxygen or glucose, directing blood flow to regions of neural activation [8]. In neonates, the myogenic response reserve of cerebral blood vessels is limited and the response to changes in PaCO_2_ is only incompletely developed. Compared to adult physiology moderate hypocapnia has little effect on CBF alteration. However, in neonates the CBF increases under conditions of lower PaO_2_ [9].

Apart from metabolic demands, CBF depends on the cerebral perfusion pressure (CPP), which is the difference between the mean arterial pressure (MAP) and ICP. Under normal conditions, CBF is kept within narrow boundaries. A constant blood flow independent of the mean arterial pressure (MAP) is necessary for a consistent supply of O_2_ and glucose and elimination of metabolites in brain cells [10]. CBF is therefore autoregulated at a constant level in response to changes in MAP: a drop in MAP will lead to cerebral vasodilation in order to keep CBF constant, and vice versa. A pathologically elevated ICP lowers CPP and thereby CBF [11,12].

For structural and functional integrity of the brain, the intracranial pressure needs to be tightly regulated. Normal ICP varies with age starting at 1–5 mmHg at birth, then continuously increasing to 8–15 mmHg in adults. At birth, the cranial sutures are open and gradually close throughout growth: the posterior fontanelle at the age of 3 to 6 months, followed by the anterior fontanelle at 10 to 18 months. Final cranial suture closure occurs in the school age.

As mentioned earlier, hypoxia and hypercarbia cause cerebral vasodilation with an increase in volume of the blood compartment. On the contrary, mild hyperventilation (paCO_2_ ≥ 30–35 mmHg) can be applied as a short-term measure to lower ICP by cerebral vasoconstriction without compromising oxygen supply in ventilated patients. Other interventions to lower ICP include perioperative steroid therapy (e.g. dexamethasone) to reduce peritumoral oedema, administration of mannitol or hypertonic saline to reduce cerebral oedema, or the insertion of an external ventricular drain (EVD) in case of obstructed flow of CSF. If the above-mentioned therapy has failed to lower ICP below 25 mmHg, a barbiturate administration should be considered, as that tends to lower ICP when standard therapeutic measures fail [12]. However, therapy with barbiturates results in an increased risk of cardiovascular instability. Close attention should be placed on maintaining adequate volemia, as well as MAP and, ultimately, CPP. Vasopressor therapy to maintain adequate CPP is often required.

Age specific treatment thresholds for elevated ICP and low CPP are not exactly defined for any underlying pathology. Recent guidelines for severe traumatic brain injury suggest treatment of ICP targeting for a threshold of <20 mmHg with a CPP target of 40–50 mmHg to ensure that a minimal pressure of 40 mmHg is not undercut [11].

## 3. Anaesthetic Perioperative Management

The clinical features of neuro-paediatric tumours are diverse and depend on the location of the tumour and surrounding structures. Tumorous tissue can directly cause focal neurological symptoms such as cranial nerve palsies, focal muscle weaknesses, hyponatremia in the setting of SIADH or hypothalamic–pituitary hormonal deficiencies. Other symptoms such as headache, lethargy, seizures, or nausea and vomiting are related to increased intracranial pressure caused by a direct mass effect of the tumour or by CSF flow obstruction resulting in hydrocephalus. In infants, sometimes the only sign of a brain tumour may be a failure to thrive. A detailed history, physical examination, review of radiological examinations and lab work are mandatory in order to minimize morbidity. Perioperative steroid therapy is administered in most cases to diminish peritumoral oedema, and usually weaned over several days after surgery.

During the preoperative assessment, children and caregivers must be approached with thoughtfulness. Often, they faced the diagnosis only a few hours earlier and are still struggling to realize its extent. The focus should centre on a personal, age-appropriate communication with the child and an empathetic conversation with the caregiver about perioperative risks and potentially relevant complications. Direct communication and collaboration with the responsible surgeon are absolutely mandatory to ensure minimal perioperative difficulties.

The setting of anaesthesia induction must be adapted to the age and cognitive development of the child. Neurologically stable children over 6 months are commonly premedicated with midazolam 0.3 mg/kg at our institution, aiming to ease anxiety. From the age of 6 months, parental presence at induction is welcomed, as it can be an excellent way to comfort both child and caregiver. The induction process needs to be discussed beforehand to maximally ensure safety aspects. Psychological consideration for the caregiver is of equal importance, as they enter an environment completely unknown to them. In neurologically stable patients lacking symptoms of elevated intracranial pressure, anaesthesia can be safely induced by sevoflurane/nitrous oxide by mask or propofol intravenously, depending on anxiety and already existing intravenous catheter. A very short use of nitrous oxide at the beginning of induction should not be a problem with neurosurgical procedures. Nitrous oxide will be avoided after induction to minimize the risk of increases in cerebral blood flow and CMRO_2_, and postoperatively, raised intracerebral pressure in pneumo-encephalus.

To facilitate laryngoscopy endotracheal intubation, neuromuscular blocking agents are administered. If pre-existing elevated, symptomatic intracranial pressure is present, a modified rapid sequence induction is necessary to minimize the risk of gastric aspiration and cerebral hypoperfusion. As intracranial hypertension is exacerbated by hypercarbia and hypoxia and children tend to have a low hypoxic tolerance, low pressure ventilation by mask is therefore mandatory until the relaxant takes effect Relaxation is continued, unless neuromonitoring is in place. In those circumstances a closed loop communication with the neuromonitoring team is of crucial importance.

At least two large bore intravenous lines and an arterial line for arterial blood pressure monitoring and for serial sampling of blood gases, electrolytes and haematocrit should be in place. If initial attempts at venous access fail, a central venous access must be considered [12]. Routine insertion of central venous access is not recommended but should be considered in patients at risk of massive bleeding and with a potential need of vasopressors to stabilise the hemodynamic status. In the case of a sitting position or overall higher risk of venous air embolism, a central venous access combined with a precordial Doppler ultrasound monitor could play a role in the detection and management of air embolism. Large bore central catheters are often too large for infants and many children and central venous pressure is not a reliable measure of preload, especially in prone position [13]. If insertion of a central venous access is mandatory, catheterization of a femoral vein is preferred, as it lacks the risk of pneumothorax and it does not interfere with cerebral venous return. Furthermore, the femoral vein catheter is intraoperatively readily accessible for the anaesthesiologist. A successive fluid therapy has the goal of maintaining normovolemia and thus hemodynamic stability. Isotonic crystalloids such as Ringer’s acetate are used as maintenance and, in children under the age of one, 1% glucose is added to reduce the risk of hypoglycaemia. Adding 1% glucose to an isotonic electrolyte solution has been shown to maintain electrolytes such as potassium and sodium within normal ranges and contributes to keeping the acid-base status within the normal physiological limits [14]. A lack of glucose supply increases the rate of lipolysis and may result in ketoacidosis, especially after prolonged preoperative starvation [15]. Hyperglycaemia, however, worsens reperfusion injuries. Therefore, blood glucose deserves close perioperative monitoring, especially when the infusion rate of the glucose containing fluid is increased above maintenance requirement or is used as intraoperative fluid therapy [14,16].

After induction, the patient’s blood pressure may decrease substantially. Countermeasures include fluid bolus, change of depth of anaesthetic or administration of vasopressors. The aim is to maintain blood pressure and heart rate within 20% of baseline and titrate depth of anaesthesia appropriately [17]. Cerebrovascular autoregulation maintains cerebral blood flow within a specific range in response to mean arterial blood pressure changes. Timely corrections are mandatory, although we still have sparce information regarding exact limits. On the other hand, it is known that neonates are very vulnerable to blood pressure decreases and have limited autoregulatory reserves [18]. Vasopressors are needed occasionally to maintain cardiovascular stability and/or to restore adequate cerebral perfusion and oxygenation. Administration of dopamine (in younger children) or norepinephrine (in older children) may be first considered. Dopamine, norepinephrine, and phenylephrine all act to increase cerebral blood flow, whereas norepinephrine proved to be most predictable [19,20].

Access to the patient is of crucial importance for both the neurosurgeon and the anaesthesiologist and depends on vigilant planning and careful patient positioning. Every position comes with its specific risks and benefits. The prone position is commonly utilized for posterior fossa surgery, alternatively the sitting position affords optimal chest wall compliance in patients with respiratory disease and obesity. The risks involved with a sitting position likely exceed its benefits; a sitting position is in our institution a rare exception only installed when absolutely mandatory for a suitable surgical access. For every patient, it is important to ensure free thoracic and abdominal wall motions. An increase in abdominal pressure may impair ventilation, lead to veno-caval compression and increase epidural venous pressure and subsequent bleeding. Lateral soft rolls are commonly used to elevate and support the lateral chest wall and pelvis [12]. If the head needs to be elevated for surgical purpose and to facilitate venous and cerebral spinal fluid drainage, awareness of an increased risk of venous air embolism (VAE) is crucial. Especially in neonates and patients with pre-existing cardiac defects, there is a risk of paradoxic emboli (right-to-left flux via cardiac lesions) [21].

The existing methods to detect VAE show varying sensitivity. Therefore, this translates to an extended incidence range of VAE, obscuring the true number of clinically relevant incidents.

Neurosurgical procedures include the highest risk for VAE. This is because the wound field is frequently located in an elevated position relative to the level of the right heart. Numerous large, non-compressible venous communications breach the dural sinuses [22]. If VAE is observed and hemodynamic stability is compromised, the surgeon should be informed immediately; he should cover the surgical field with saline-soaked dressings and the patient should be placed in Trendelenburg position in order to stop, or at least limit, further entrainment of air and to improve cerebral perfusion. Left lateral positioning might be difficult to achieve in patients undergoing craniotomy surgery and has been suggested to be ineffective [23]. Moreover, 100% high-flow oxygen should be administered to accelerate elimination of nitrogen. The success rate of air aspiration in the right atrium during VAE is very low and there is no data to support emergent catheter insertion during an acute setting of hemodynamic compromise.

Extreme head flexion or rotation of the head must be performed with care, as it may cause brainstem compression or impair venous return via the jugular vein. Whatever position is applied, great care must be given to the proper fixation of the endotracheal tube, the positioning and securing of the eyes and the padding of pressure points (Figure 2, Figure 3 and Figure 4).

Anaesthesia for paediatric craniotomy surgery is usually maintained with either volatile agents such as sevoflurane with a minimum alveolar concentration of ≤1 MAC, or with total iv anaesthesia. However, in our opinion, propofol has some advantages over volatile agents. Compared to sevoflurane, the use of propofol has been associated with lower intracranial pressure and cerebral swelling at the opening of the dura and higher mean arterial blood pressure and cerebral perfusion pressure [24]. This study claimed improved operating conditions when using propofol instead of sevoflurane. Regional cerebral blood flow and regional cerebral blood volume appeared to be better controlled under propofol compared with sevoflurane-based anaesthesia. Further benefits include the anti-convulsive properties of propofol, while the use of sevoflurane in high concentrations may entail epileptic potentials. However, prolonged use of propofol should be considered carefully in younger children, as there is always the potential of a propofol infusion syndrome, even more so in critically ill patients suffering from neurologic diseases.

Paediatric patients undergoing craniotomies may suffer significant blood loss. The amount of blood loss varies depending on the size, vascularisation, histological type, location, and proximity to major blood vessels of the tumour. Blood loss can be difficult to estimate, and its assessment needs some experience. Apart from obvious losses, a constant oozing onto surgical drapes and irrigation fluid must be considered.

There are no specific transfusion recommendations for paediatric brain surgery, however, taking into account the newest paediatric intensive care guidelines [25], reasonable goals include haemoglobin > 7–10 g/dL and thrombocytes > 100,000/μL, respectively. In the case of significant bleeding, thromboelastography can not only help to identify specific problems such as functional lack of fibrinogen or hyperfibrinolysis, and therefore help to trigger optimal therapy, but also allow immediate evaluation of treatment effects (point-of-care analysis).

Maintaining normovolemia does not only mitigate hemodynamic fluctuation, but also minimizes the risk of venous air embolism (VAE).

Based on the specific location of the lesion, specific perioperative complications need to be anticipated. The majority of paediatric brain tumours are located in the posterior fossa exerting a mass effect and often obstructing cerebrospinal fluid flow. Brain stem tumours may impinge on the respiratory control centre and cranial nuclei, potentially resulting in hypertension and tachycardia or bradycardia and vocal cord paralysis, depending on the involved nuclei. These structures are also vulnerable to surgical manipulations. A vigilant awareness of the specific situation is imperative for deliberate monitoring and intraoperative maintenance of anaesthesia.

After classic craniotomies, neurologic function is assessed clinically after patients regain consciousness. Therefore, the goal is a rapid emergence from anaesthesia in order to allow immediate neurologic examination and to detect complications early. Interestingly, recent studies show no difference after propofol or sevoflurane-based anaesthesia when combined with either fentanyl or remifentanil in terms of recovery and cognitive function [26].

## 4. Postoperative Management

Children undergoing posterior fossa craniotomies, after ischemia or oedema of the brainstem, and children with Chiari malformation may be at risk for postoperative respiratory depression or apnoea. Continuous observation in an intensive care setting for the first postoperative hours is advisable, aiming to rapidly observe and treat respiratory complications. After surgery in the hypothalamic region and pituitary gland, imbalances of electrolytes occur frequently (e.g., diabetes insipidus) and need to be treated with iv fluids and hormones. Postoperative nausea and vomiting should be prevented and treated firmly since vomiting may give rise to increases in intracranial pressure.

The extent of pain after craniotomy is associated with the length of the surgical procedure, older age and infratentorial surgery. Children treated with a multimodal analgesia regime including strong opioids experience lower pain scores [27]. Acetaminophen is routinely given intraoperatively, as are nonsteroidal anti-inflammatories (NSAIDs) at the latest after 24 h after surgery. The use of NSAIDs is restricted in many institutions because the inhibition of cyclooxygenase-1 (COX-1) results in inhibition of prostaglandins, altering platelet function within normal limits in healthy patients. A recent retrospective study [28] investigated the effects of a perioperative short-term use of ketorolac after neurosurgical procedures and observed no significantly increased bleeding risk. Acetaminophen is considered a weak analgesic drug. However, in combination with NSAIDs in a multimodal pain therapy regime it is very beneficial and includes opioid sparing effects [29,30]. Opioids are often the main analgesic drug used after craniotomies. Nonetheless, concerns exist because of their sedating properties and side effects such as nausea and vomiting. However, a recent study showed that iv opioids neither altered the neurologic exam nor increased the frequency of neurologic exams regardless of the route of administration [31]. When used within a multimodal strategy, opioid consumption and side effects can be reduced [32]. Morphine, fentanyl, and hydromorphone are all strong opioids suitable for use after craniotomies and can all be administered iv depending on age, either by a continuous infusion in pre-school aged or disabled children, or by a patient-controlled anaesthesia pump (PCA) allowing the patient restricted control over their pain medication within tightly assigned parameters. To smoothen the transition to postoperative therapy, a morphine bolus at a dosing between 50–100 μg/kg combined with the first dose of ketorolac 0.5 mg/kg may be administered after closure of the dura. The use of co-analgesics such as ketamine and dexmedetomidine to reduce opioid needs may be considered. Ketamine, an N-methyl-d-aspartate (NMDA) receptor antagonist (primarily an anaesthetic), is, when used in sub-anaesthetic doses, a potent analgetic. Ketamine has been restricted in craniotomy surgery due to its effect in increasing intracranial pressure and cerebral blood flow. When used postoperatively, added directly to the PCA-pump in a dose of 10–20 μg/kg/dose combined with a similar morphine bolus, it improves postoperative analgesia while reducing opioid requirements and side effects [33,34]. Dexmedetomidine, a newer α2-receptor agonist, has intraoperative opioid and anaesthetic sparing effects. It is suggested, that through inhibiting central neurotransmitter release it could reduce neuronal damage [35]. An intraoperative infusion at a dosing of 0.5 μg/kg/h reduces pain scores and perioperative opioid consumption [36]. Thus, multimodal analgesia benefits from the combination of a wide range of analgetic drugs with different qualities to treat postoperative pain.

## 5. Conclusions

Aiming to minimize perioperative morbidity in neurosurgical paediatric patients, an age-dependent detailed preoperative evaluation, open communication between the members of the surgical and the anaesthesia teams, and advances in subspecialty training in paediatric neurosurgery and paediatric anaesthesia are essential. Understanding neurological principles in these very vulnerable patients, careful positioning, comprehensive clinical perioperative monitoring, and vigilant attention may facilitate timely correction of potentially devastating complications and are key for successful perioperative management.

## Figures and Tables

**Figure 1 children-09-01539-f001:**
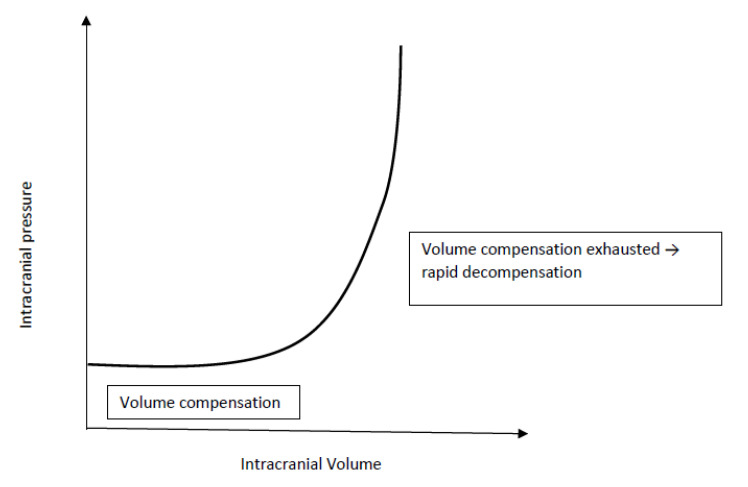
Intracranial compliance curve.

**Figure 2 children-09-01539-f002:**
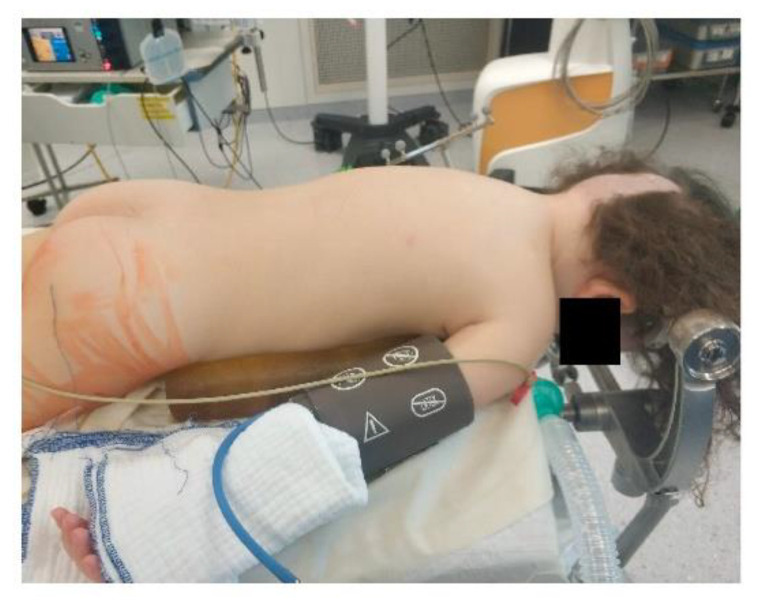
Prone positioning with lateral soft rolls to elevate and support lateral chest wall, tubus fixation.

**Figure 3 children-09-01539-f003:**
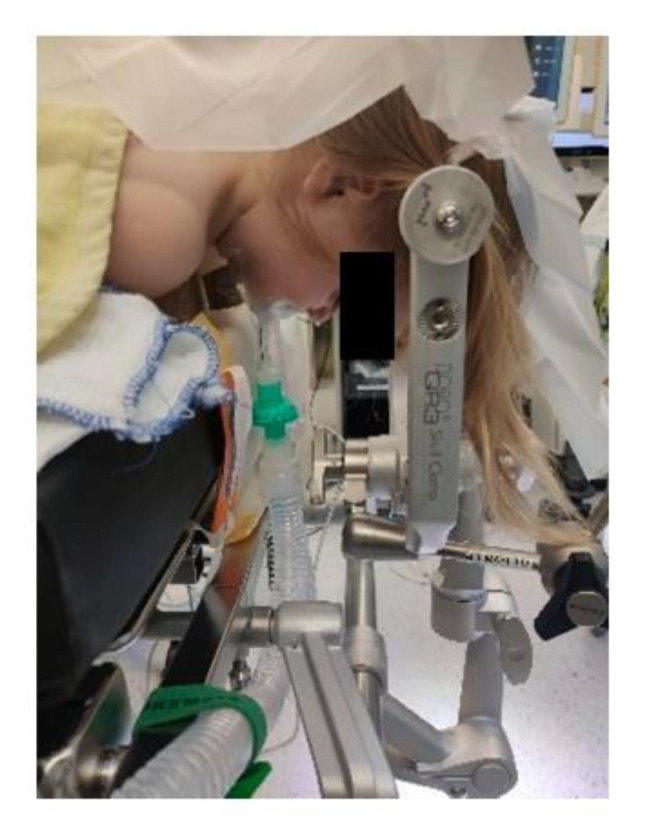
Prone positioning with lateral soft rolls to elevate and support lateral chest wall, tubus fixation.

**Figure 4 children-09-01539-f004:**
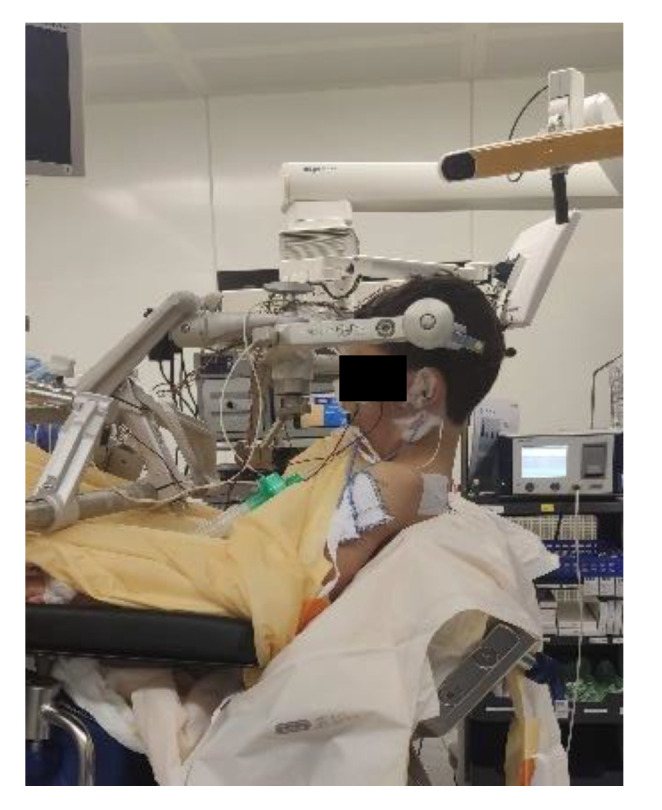
Sitting positioning [12] (Pictures reprinted/adapted with permission from Jehuda Soleman, MD).

**Table 1 children-09-01539-t001:** Cerebral blood flow for different age groups [6,7].

Age	Cerebral Blood Flow (mL/100g/min)
Premature neonate	12–20
Full term neonate	23–40
6 months to 3 years	90
3–12 years	100
Adult	50

## Data Availability

Not applicable.

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
