# Peer review of "Special Anaesthetic Considerations for Brain Tumour Surgery in Children"

_children, 2022, doi:10.3390/children9101539_

Round 1

Reviewer 1 Report

A very brief review of anesthetic considerations for pediatric neurosurgery patients. Pediatric specific physiological parameters were informative. However, the description of anesthetic concerns mainly include a narrative description of general practice in the author's institution, rather than an exhaustive review of available evidence from pediatric neurosurgical anesthetic literature. 

what is the current evidence for using 1% glucose for patients under the age of 1?

what is the current evidence on the incidence of air embolism in pediatric neurosurgical procedures?

what is the role and evidence for sedation medications other than opioids such as dexmedetomidine?

What is specific in pediatric age group for neuromonitoring? (ex- when does the myelination process complete or at what age can you expect good MEP/SSEP signals)

Please consider adding sub-sections addressing different neurosurgical procedures: ex- craniofacial reconstructions ( has specific concerns with blood management, airway management), chiari (might need valasalva intraop, prolonged sedation in ICU might be needed in pediatric patients to keep patients flat for 48-72 hrs in certain dural repairs), tethered cord release (EMG monitoring)

Author Response

Response to the reviewers

Dear Editor

We thank the reviewers for their valuable comments and suggestions. The aim of this review, is a condensed description of anaesthesia for brain tumour surgery in children.

Reviewer 1:

  • Point 1 and 2 are addressed in the revised manuscript
  • Not clear what is meant with dexmedetomidine for sedation instead of opioids. If question is sedation for awake craniotomy, this is not the scope of this review. Postoperative sedation for dura reconstruction is not done in our institution and not known as a therapy by our neurosurgeons. Adding subsections for different neurosurgical procedures is not the scope of this review, as we were asked to write about anaesthesia for brain tumours in children.
  • Neurophysiological monitoring interferes with anaesthetic considerations. However, this is a distinct topic and is not the scope of this review.

Reviewer 2 Report

This is a review article on the anesthesia management for pediatric intracranial tumors.  The article appropriately summarizes it’s goals and achieves them in the narrative of the article.  They are well organized in related to concepts of physiology, preoperative, intraoperative, and postoperative management.  The reference articles are appropriately cited and do not see any obvious errors of logic.

There is considerable (welcomed) detail in the pathophysiology of CBF, metabolism, and ICP. 

It is worth considering adding other treatments for ICP including barbiturates and Lasix.  Although not frequently administered, I find review articles as opportunities to remember alternative treatments. 

Also consider a brief review of treatment of VAE.

Additionally, consider adding considerations to the type of vasopressors and vasodilators as related to cerebral physiology. 

Finally, I found the following sentence confusing:

In neonates, the myogenic response reserve of cerebral blood vessels is limited and the response to changes in PaCO2 is only incompletely developed. However, the sensitivity to decreases in PaO2 is relatively higher (9)

It was difficult for me to understand the message.  I think you are referring to the diminished autoregulatory response of neonates but think it can be clearer. 

Author Response

Response to the reviewers

Dear Editor

We thank the reviewers for their valuable comments and suggestions. The aim of this review, is a condensed description of anaesthesia for brain tumour surgery in children.

Reviewer 2:

  • Barbiturates as a treatment option of increased ICP is now included in the revised manuscript. The use of Lasix for attempt to lower the ICP is not common at all in our practice and is not a treatment option even in comprehensive contemporary guidelines.
  • Treatment of VAE and the use of vasopressors are now included.
  • Sentence in question is adapted.

Best Regards,

Sandra Jeker

Round 2

Reviewer 1 Report

Thank you for addressing the comments. The article provides a comprehensive description of a single institutional practice of anesthetic management of neurosurgical procedures. 

Author Response

Thank you for your comment, time and thoughtful input.

Best regards